

# Surrogate models for unsteady aerodynamics using non-intrusive Polynomial Chaos Expansions

Rad Haghi[1] and Curran Crawford[1]

[1]Institute for Integrated Energy Systems, University of Victoria, British Columbia, Canada

**Correspondence:** Rad Haghi (rhaghi@uvic.ca)

**Abstract.** In common industrial practice based on IEC standards, wind turbine simulations are computed in the time domain for each mean wind speed bin using six unsteady wind seeds. Different software such as FAST, Balded or HAWC2 can be used to this purpose, to capture the unsteadiness and uncertainties of the wind in the simulations. The statistics of these simulations are extracted and used to calculate fatigue and extreme loads on the wind turbine components. Having only six seeds does not guarantee an accurate estimation of the overall statistics. One solution might be running more seeds; however, this will increase the computation cost. Moreover, to move beyond Blade Element Momentum (BEM)-based tools toward vortex/potential flow formulations, a reduction in the computational cost associated with the unsteady flow and uncertainty handling is required. This study illustrates the stationary character of the unsteady aerodynamic statistics based on the standard turbulence models. Afterwards, we propose a non-intrusive Polynomial Chaos Expansion (PCE) to build a surrogate model of the loads' statistics at each time step, to estimate the statistics more accurately and efficiently.

## 1 Introduction

The process of calculating loads on wind turbine components is one of the core parts of wind turbine aerodynamic and structural design and optimization. In the last few decades, international organizations have developed different aeroelastic codes such as FAST (Jonkman et al., 2005), BLADED (Bossanyi, 2003) and HAWC2 (Larsen and Hansen, 2007) to calculate load time series based on the standardized or site-specific environmental conditions accurately. Engineers and researchers use simulation output statistics to calculate extreme and fatigue loads wind turbine structure and estimate the unsteady power. To take into account the randomness in the unsteady wind, according to IEC standards (IEC 61400-1), the simulation process must use a Monte Carlo (MC) method. Therefore, the simulation setup should include a limited number of seeds for generating wind speed time series of $600s$. In normal practice, for each mean wind speed, at least six different seeded unsteady wind time series is required as the minimum to take into account the uncertainties. This limited number of deterministic but unsteady simulations do not yield an accurate and smooth estimation of the statistics. Gradient-based optimization algorithms may not deal with these non-smooth statistics well. One option to solve this problem is running more seeds, which will increase the computational cost. The increase in the computational cost will play a more important role in our decision making if we want to move towards vortex (van Garrel, 2003) and potential flow codes for load calculations, as they require more computation





resources inherently. An alternative approach is to use a surrogate model that can provide us with a smooth set of statistics based on a limited number of simulations.

The origin of the *surrogate model* lies in Uncertainity Quantification (UQ) analysis (Sudret, 2007). There are many uncertainty quantification implementations in wind energy. More specifically, in wind farm load estimation or optimization, surrogate models show much potential. Many researchers have investigated these potentials (Dimitrov et al., 2018; Schröder et al., 2018;

Dimitrov, 2019; Ashuri et al., 2016). However, very few have looked at building a surrogate model of the aerodynamic model of wind turbine. Fluck and Crawford showed an initial attempt to build a surrogate model based on intrusive Polynomial Chaos Expansion (PCE) on simple lifting line and BEM models (Fluck, Manuel and Crawford, Curran, 2016; Fluck and Crawford, 2018). As they quickly faced with *curse of dimensionality*, they showed it is possible to reduce the number of random variables in Veers' unsteady wind model significantly. Afterwards, they used this reduced dimension wind model to propagate

stochasticity thorough a simple lifting line (Fluck and Crawford, 2016) or BEM (Fluck and Crawford, 2018) model. However, with intrusive PCE it is necessary to change the model fundamentally to be able to incorporate the random variables (Sudret, 2007). For a simple model, this might work, but when we want to utilize commercially available aeroelastic codes, this will be challenging or even impossible.

This paper is an initial attempt to build a non-intrusive PCE surrogate model (Sudret, 2007) on a deterministic aerodynamic

model. The motivation is building a surrogate model based on a few numbers of simulations to be able to estimate the statistics of aerodynamic model output at each time step of the time series quickly and accurately. As the surrogate models are inherently cheap to run, we take this surrogate model through a Monte Carlo simulation (MCs) for a large number of times. The input of these MCs is the samples which are drawn from the uniform random variables. An unsteady wind generator uses the same random variable distribution. The output of the surrogate model provides us with the aerodynamic model output distribution

quickly. This process is presented in Figure 1 schematically. By this method, we can reduce the computational cost and time for the aerodynamic simulation, without compromising the validity of the results. One can interpret this model as a tool to map the input distribution (in this case, an uniform distribution of random seeds-phases) to the output distribution (in this case, a normal distribution of $\Gamma$).

As fitting a surrogate model of each time step of a $600s$ times series is computationally expensive and redundant to current

practice, we start by showing that the aerodynamic simulation results based on Veers reduced model (Fluck, M. and Crawford, C., 2017) statistically converges. This means that the statistical properties of the unsteady process are constant in time. Therefore, by keeping the computational effort the same, it is possible to run more simulations while shortening the length of simulations. More simulations with the same computational effort give us the chance to fit higher degree PCEs, which provides a more accurate estimation of the statistics. We build four different PCEs surrogate models, with four different polynomial

degrees. These surrogate models have been used for MCs for a large number of runs (cheaply). The results of the MC runs of the surrogate models are compared with $48000$ unsteady wind aerodynamic simulations. We compare the results by looking at the circulation distribution from both the deterministic model and the surrogate model.



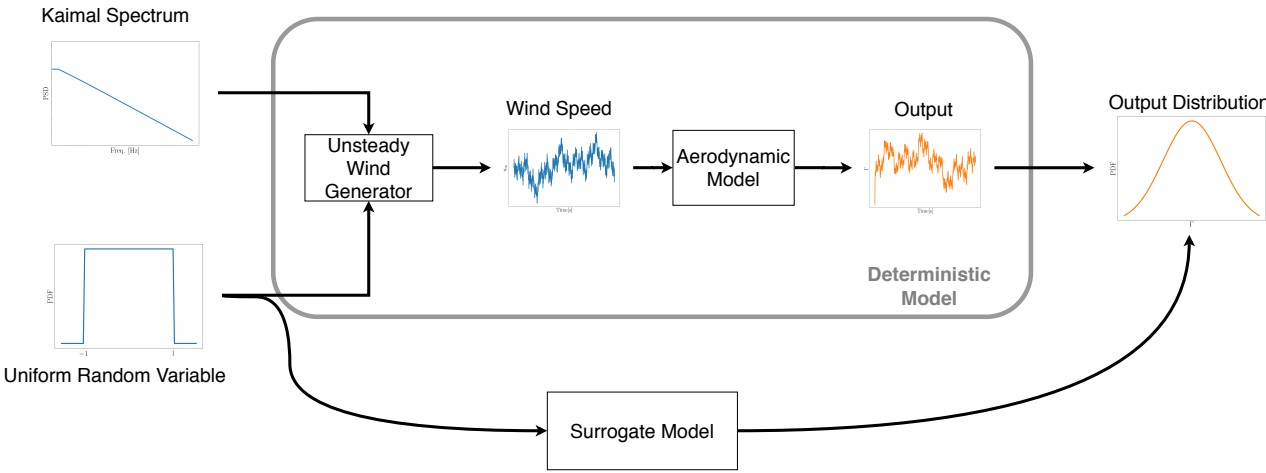

**Figure 1.** The common deterministic process of aerodynamic modeling vs the suggested surrogate model method schematic flow chart.

## 2 Methodology

This section starts by providing an overview of the basic wind environment and statistical elements that we use in this study.
Afterwards, we explain in detail the simulation and PCE method that we use in this study.

### 2.1 Reduced Veers unsteady wind model

One famous unsteady wind model in the wind turbine community is the Veers model (Veers, 1988). The history of the model goes back to the late 80s' and has a long record in wind turbine load calculation practice. Veers' model for the unsteady wind is the method to generate turbulent boxes, commonly implemented in TurbSim (Jonkman, 2009). The method is briefly explained
in Fluck, M. and Crawford, C. (2017) and extensively in Veers (1988). To make the unsteady wind, this method roughly needs $5 \times 10^4$ random variables based on typical wind frequency bins for each phase $\phi_j$. For reasons that we will explain in the next sections, this number of random variables makes building a PCE surrogate model almost impossible. Fluck and Crawford (2016) showed that using only ten uniformly distributed independent random variables with ten frequencies logarithmically sampled from the Kaimal spectrum (Veers, 1988) are enough for building unsteady wind time series. This *Reduced Veers'*
*model* generated unsteady wind that can capture the same level of randomness and probability distribution as the full model. In this study, we used this reduced Veers model to generate unsteady wind time series.

### 2.2 Statistical convergence

For this study, we want to investigate if the probability distribution of $n$ wind seeds and aerodynamic simulation results at each time step are similar or not. In other words, we want to know if the statistical properties of the output at each time step converge.

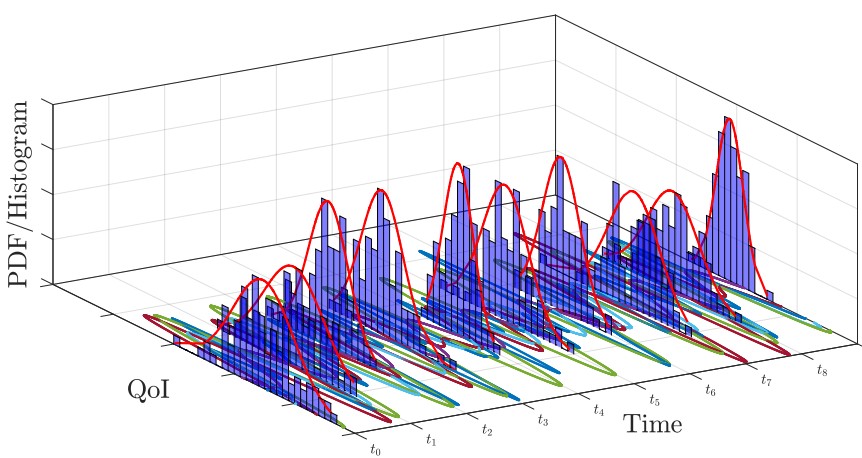

**Figure 2.** A schematic drawing presenting possible distributions at each time step based on a set of time trajectories for a Quantity of Interest (QoI)

Figure 2 presents a generic example of distributions (histogram fits) at each time step for a set of random processes. The figure shows a schematic plot; therefore, histograms and fitted distributions *do not* represent the plotted trajectories. To show that the combination of $n$ seeds is statistically convergent, we need a metric to quantify the difference between the distributions and each time step. There are different metrics for this purpose (Basu et al., 2011); for this study, we use *Hellinger distance* (Hellinger, 1909) as a metric. The Hellinger distance is a metric to quantify how two normalized probability distributions are similar to each other. The distance is zero if they are the same and reaches to one when one of the probability distributions is zero uniformly. The Hellinger distance for two *discrete* probability distributions of $P$ and $Q$, where they are both normalized and have an equal number of bins can be formulated as:

$$H(P,Q) = \frac{1}{\sqrt{2}}\sqrt{\sum_{i=1}^{k}(\sqrt{p_i} - \sqrt{q_i})^2} \tag{1}$$

In Eq.(1), $p_i$ and $q_i$ are the normalized probabilities for $P$ and $Q$ at every bin.

### 2.2.1 Aerodynamic model

For the sake of simplicity, the aerodynamic model in this study is a simple rectangular lifting surface (wing) in an unsteady flow $u_\infty$ assumed constant across the span and one dimension flow, i.e. $\alpha_g$ incidence angle constant. This is modelled using a single Prandtl lifting line, with shed wake elements to take into account the previous time step wake effects on the lifting line



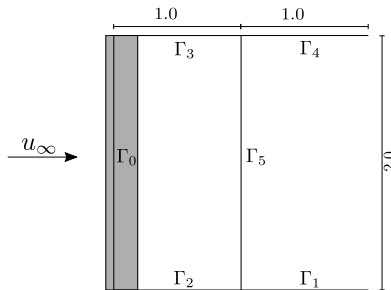

**Figure 3.** Unsteady lifting line model. Dimensions are in meters.

circulation (Figure 3) along with a single set of trailed vorticies. Again for simplicity, the wake elements are fixed geometrically,

and a single spanwise element is used to represent the wing. Relaxing this constraint into a free wake model, and increasing the number of wake elements will not change the method introduced here. However, it increases the number of unknowns and makes the aerodynamic model simulations more expensive. For a more detailed explanation of lifting line theory, see Prandtl (1919) or the more recent work of Phillips and Snyder (2000). This model assumes a small angle of attack $\alpha_g$ and small induced velocities $w$. Therefore, one can claim the modelled lifting surface has linear airfoil properties (Anderson and Hughes, 2009).

The model formulation is taken directly from Fluck and Crawford (2016):

$$\Gamma_0(t_n) = \frac{A.u_\infty(t_n) + C.\Gamma_0(t_{n-1})}{1 - B} \tag{2}$$

where:

$$A = c\pi\alpha_g$$
$$B = (G_1 + G_2 + G_3 - G_5) \tag{3}$$
$$C = (G_1 + G_4 + G_5)$$

Eq. (2) provides the ability to calculate the bound circulation (and thus loads) $\Gamma_0(t_n)$ at each time step $t_n$. This is based on the free-stream wind $u_\infty$ and the circulation at the previous time steps $\Gamma_0(t_n - 1)$. In Equation set (3), $c$ is the chord, $\alpha_g$ is the angle of attack and $G_i$ is Biot-Sawart influence term from each vortex at the middle of the blade element (Phillips and Snyder,

2000). By having a $u_\infty$ time series in hand from *TurbSim* of any other unsteady wind generator, Eq. (2) can be solved using general numerical methods. The previous circulations advect out of the model in 1 time step. Initial transient in the simulations are quickly advected out of the results time history. The mathematical model in Eq. (2) is an *Autoregressive* order one model. Therefore, if the input is a stationary process, the output of the model is stationary also, based on this order and type of model. The numerical experiment in section 3 will validate this analytic prediction and explore the statistical convergence to that limit.





## 2.3 Polynomial Chaos Expansion

Uncertainty propagation of mathematical models has been the subject of many studies in the last thirty years. One method to propagate uncertainty is using models or surrogate models. A surrogate model is a cheap-to-run approximation of the actual model (Kim and Boukouvala, 2019). Among the surrogate models, Polynomial Chaos Expansion (PCE) has gained attention especially after the work of Ghanem and Spanos (2003) and Xiu and Karniadakis (2003). PCE is a method that can describe the uncertainty in the input variable, and propagate it through the model using a basis of polynomials. This way, the uncertainty can be propagated through the model with a limited number of simulations (Tyson et al., 2015). In other words, PCE tries to estimate the response of a mathematical or numerical model based on a series of orthogonal polynomials depending on random variable $\boldsymbol{\xi}$. The solution is expanded and described in stochastic space spanned by $\boldsymbol{\xi}$ and the associated polynomial basis set.

The main reasons to use PCE instead of other surrogate model methods are: a) with minimum computational effort one can extract statistical properties from PCEs; b) PCEs are easy to integrate into deterministic linear and nonlinear mathematical models; c) One can build PCE surrogate model by treating the model as a black box (Kaintura et al., 2018; Sudret, 2015).

In order to make it easier to explain, lets assume $Y(t) = \mathrm{M}(t, \boldsymbol{\xi})$ where $t$ is time and $\boldsymbol{\xi}$ is the random variable vector, $\mathrm{M}(t, \boldsymbol{\xi})$ is our deterministic mathematical or numerical model and $Y(t)$ is the output of the model. Therefore, the stochastic output of the model $Y(t, \boldsymbol{\xi})$ can be expanded as:

$$Y(t, \boldsymbol{\xi}) = \sum_{i=1}^{\infty} y_i(t) \Psi_i(\boldsymbol{\xi}) \tag{4}$$

where $y_i(t)$ are PCE coefficients at each time step and $\psi_i(\boldsymbol{\xi})$ is a member of an orthogonal polynomial class. These polynomials are orthogonal with respect to the probability space of random variable $\boldsymbol{\xi}$. The selection of polynomial type is a function of the probability distribution on the random variable $\boldsymbol{\xi}$. For example, if a random variable $\boldsymbol{\xi}$ has a normal distribution, then a Hermite polynomial is selected (Xiu and Karniadakis, 2002). The polynomials do not necessarily need to be selected from the specific family of polynomials as long as they are orthogonal polynomials. For instance, Fluck and Crawford (2018) showed exponentials components work at best for their purposes. As the randomness in this study comes in the form of a uniform distribution for $\phi_j$, the surrogate model is based on the Legendre polynomials (Xiu and Karniadakis, 2003). In practice, the PCE summation in eq (4) is truncated at a reasonably high order $p$. The main objective of the expansion in eq (4) is finding the coefficients $y_i(t)$. There are two main approaches to solve this problem;

- the *intrusive* approach where the model is projected on the orthogonal polynomials using Galerkin projection (Ghanem et al., 2017). This approach requires building a detailed stochastic model from the deterministic model governing equations.

- the *non-intrusive* approach allows calculating the PCE coefficients from a series of deterministic model evaluations. This approach considers the model as a black box and does not require any model modification (Sudret, 2007; Eldred et al., 2008). There are two sub-categories to calculate the coefficients, namely *simulation methods* and *quadrature*



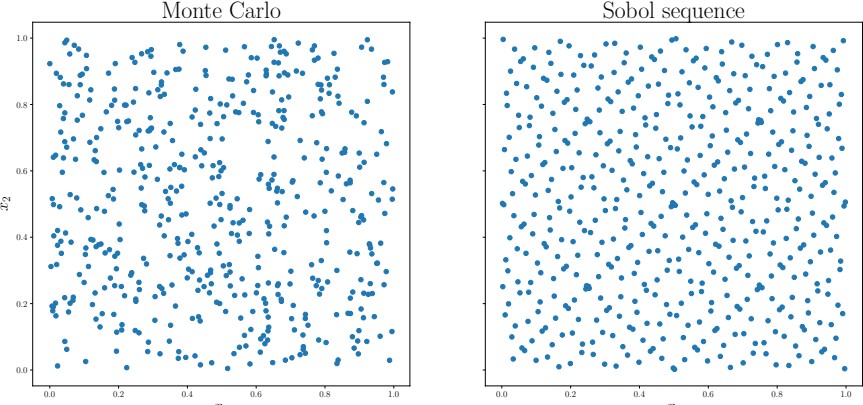

**Figure 4.** Two sampling methods drawing 512 samples from two uniform distributions using MC and Sobol methods

*methods* (Sudret, 2007).The presented work use *simulation methods* from the *non-intrusive* category to calculate the PCE coefficients.

The intrusive approach was used by Fluck and Crawford (2016, 2018) to build a surrogate model on a lifting line and BEM models. In this study, we used the non-intrusive approach to calculate the PCE coefficients. The surrogate model in
this work is built employing the Python toolbox Chaospy (Feinberg and Langtangen, 2015). Chaospy is a numerical tool for uncertainty quantification using different methods, including PCE. For this study, we used *point collocation* method to calculate the coefficients. This method has been explained well in the literature. The interested reader is referred to (Xiu et al. (2002),Ghanem and Spanos (2003),Sudret (2007)).

### 2.4 Random variable sampling

The randomness in the generated unsteady wind comes from the ten random variables in $\boldsymbol{\xi}$ described frequency components' phases $\phi_j$ in the reduced Veers model (Fluck, M. and Crawford, C., 2017). Based on the Veers methods (Veers, 1988) and in TurbSim (Jonkman, 2009), the employed sampling method is a pseudo-Random Number Generator (pRNG) which is the basis of MCs. However, the problem with this way of sampling for MCs is the lack of control on the random variables' domain as it may fill some voids in the domain, and may leave some of the empty (Niederreiter, 1992). Therefore, for the same reason,
the random domain may not be filled evenly. For this study, a low discrepancy Quasi Monte Carlo (QMC) sampling method, namely the Sobol sequence (Sobol', 1967) is used to draw samples from the random variables. A customized random wind generator based on the reduced Veers model used these samples to generate unsteady wind. The difference between these two sampling methods can be visualized. In Fig. 4, the difference is illustrated by sampling two random variables of $x_1$ and $x_2$ each uniform distribution for 512 sample points. The Sobol method distributes the points relatively uniformly over the full space.





| Polynomial Degree | Number of Coefficients + 1 | No of Required simulations | Sims. Length to fit PCEs [s] |
|---|---|---|---|
| 1 | 11 | 22 | 163.6 |
| 2 | 66 | 132 | 27.3 |
| 3 | 268 | 536 | 6.7 |
| 4 | 1001 | 2002 | 1.8 |
| 5 | 3003 | 6006 | 0.6 |

**Table 1.** The number of coefficients and required data points to calculate the coefficients for 10 random variables and collocation method. This number of coefficients should be calculated for every time step. The last column demonstrates the simulation length for the fitted PCEs as explained in Section 3.

## 2.5 Approach

This study starts with running an extensive set of simulations based on the reduced Veers model and aerodynamic model explained previously. As we have a large number of simulations, we can show that the circulation statistics with time are not changing significantly, and therefore the process statistics converge as analytically expected. Knowing the process statistics converge, we conclude only building a single surrogate model, i.e. for a single time-step or a few ones, would suffice for our purpose. The accuracy of the PCEs depends on the polynomial degree. However, an increase in the polynomial degree pushes the problem further toward the *curse of dimensionality*. The number of required coefficients to build the surrogate model, and the required number of simulations are presented in Table 1. The table shows we need a large number of simulations to build an accurate PCE. Also, we want to limit the computational cost to 6 times $600s$ simulations (in total $3600s$) to be competitive with the common practice in wind turbine aerodynamic simulation. The combination of these requirements leads to a large number of short simulations instead of a few long ones. To make this trade-off fair, we kept the cumulative length of the simulation at 3600 seconds. This means that as the length of the simulations decreases, the number of simulations increases. Sobol sampling is the base of the simulations setup. For every set of the required number of simulations in Table 1, the random phases are drawn independently from the rest of the sets. For example, for the second row of Table 1, when we need 132 simulations, 132 unique samples of $\xi$ drawn from the random domain. These $\xi$ have not been used for other simulation sets. By having a large number of data points at each time step, we built a few surrogate models in time and compared the results with the main database of the simulations. For the sake of accuracy, in this study, we do not build any surrogate model based on 1st-degree polynomials.

Calling back Eq. (2), we recognized the model is an autoregressive model. For a stationary input, the sample statistics of output converges at the rate of $1/\sqrt{n}$, where $n$ is the number of data points (in this case, $48000$ data points for each sample). Consequently, it is possible to *estimate* the statistical parameters of the output distribution by different methods; such as, using the maximum likelihood estimator Rao (2008). Then a question that arises is why we go through the complication of building a surrogate model. The goal of the research we present here is building a surrogate model of an aerodynamic model whether





the aerodynamic model is simple or complex. We choose this simple aerodynamic model for the ease of simulations; hence, the simplicity of the aerodynamic model does not compromise the validity of the method.

## 3   Results

As mentioned before, in Section 2.5, we started by running a broad set of reference simulations. For this case, we ran $48000$ simulations for a $5m/s$ wind speed and turbulence intensity of $0.16$. The wind generator code took $48000$ samples from a $10$ dimensional uniform distribution domain based on the Sobol sampling method. Each sample is a $10$ by $1$ vector of $\boldsymbol{\xi}_j$, and we have $48000$ of them. $48000$ wind speed histories were generated and simulations on the aerodynamic model run with a time step of $0.1s$ for $600s$ (in total $6000$ time steps per simulation). This simulation setup builds a database for the investigation, and to show that the process distributions at each time step changes are insignificant. The histogram at each time step was calculated and normalized. Afterwards, using the Hellinger distance formulation, the distance between each histogram to the other histograms ($5999$ other histograms) was calculated and stored in a matrix. Each row of this matrix shows the difference to the histogram at one-time step compared to the other ones. Therefore, this is a symmetric matrix with zeros on the diagonal. What is important is the maximum of all of the data in the matrix; in Figure 5, we show the max of the Hellinger distance at each time step. The Hellinger distance is a normalized metrics and the distances shown in the percentage. The plot shows a comparison of all the 18 million possible combinations to calculate the Hellinger distance; the difference between the distributions does not exceed $2.42\%$. Therefore, we can conclude that building a surrogate model on a limited number of time steps is enough, and it is not necessary to build a surrogate model on every time step as predicted by the aerodynamic model form.

Referring to the varying number of simulations with respect to PCE degree in Table 1, the first $100s$ of the simulations is plotted in Figure 6. These simulations are input for building the surrogate models. The number of samples that are drawn from the $10$ dimensional random space is equal to the number of simulations. The employed sampling method is Sobol. This manner is similar to the way that we run our reference case with $48000$ simulations.

Since the statistics of the simulations are essential for this study, for the presented simulations in Fig 6, one can calculate mean and standard deviation and their propagation in time. Figure 7 shows these values for the first $100s$ of the simulations and compares it with the reference case mean and standard deviation propagation in time. Figure 7 shows by visual inspection the mean value for these number of simulations is close to the reference case; however, the standard deviation is not yet convergent.

As it may not be clear, the plots in Figure 7, Figures 8 and 9 present the mean and standard deviation of the reduced set of simulations histograms in comparison with the reference case. In other words, these plots show the histograms of the 6000 time steps mean and standard deviations. In Figure 7, it seems the mean and standard deviation values are very similar to the reference case; they have a wider distribution.

Referring to the discussion at the beginning of this section, for the reference case, the changes in statistical properties at each time step are minimal (Figure 5). Therefore, a few accurate surrogate models would suffice to emulate the aerodynamic simulations. By this assumption, building surrogate models is more feasible from a computational cost point of view. As

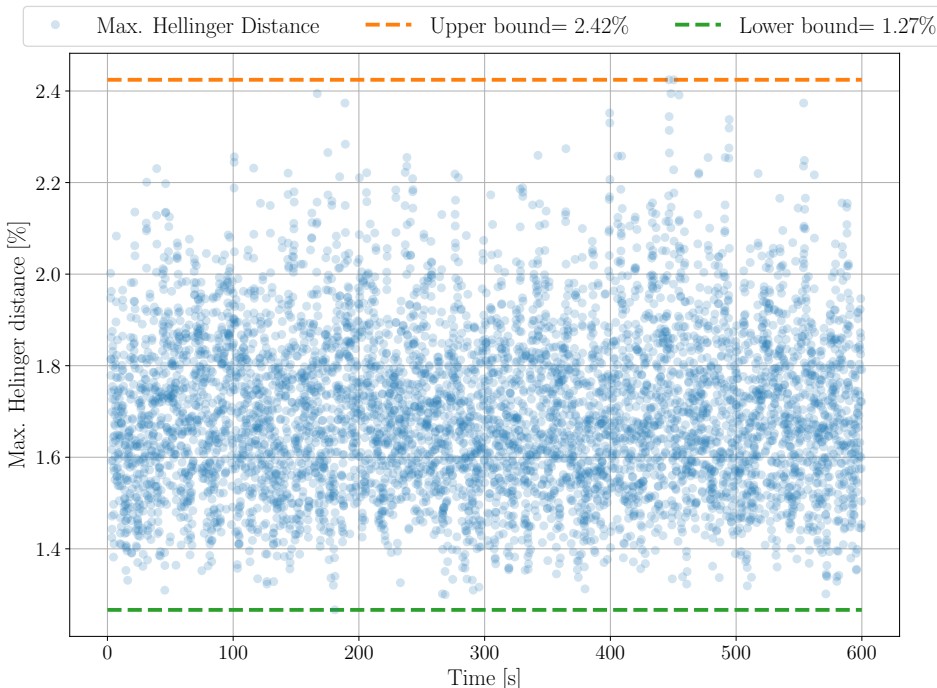

**Figure 5.** Maximum of Hellinger distance at each time step. The upper and lower bounds for the the *maximum* of the Hellinger distance are indicated.

explained in Section 2.5, we fit surrogate models on a large number of short simulations instead of a few long ones. The number of simulations is the same as mentioned in Table 1, and the length of the simulations that the surrogate model is built in the last column. We used those shorter simulation lengths to build the surrogate model, as visualized in Figure 10. The vertical line shows where the cumulative simulation length is $3600s$. Specifically, the vertical line shows the cut off for PCE building.

Using Chaospy (Feinberg and Langtangen, 2015), we fit PCEs on the length of the simulations indicated in Table 1. Namely, for every time step, we create a surrogate model based on the number of simulations and the polynomial degree by calculating the coefficients of the polynomials (Eq. (4)). In Figure 10, the calculated mean value and standard deviation from the reference case are compared with the estimated ones from the PCEs computed from the PCE coefficients directly. The results in Figure 10 show the PCEs fit for four polynomial degrees; $P$ on each plot indicate the polynomial degree. As the polynomial degree

increases, the fit to the reference case improves, which is expected. Although it was not necessary, for polynomial degrees 3 to 5, the PCEs are fit to the whole $10s$ of the simulations to have an acceptable sample size for comparison.

     The goal of building surrogate models is to create an emulator that we can run quickly and can provide statistics for the simulations without actually running the simulations. To test the accuracy of the surrogate models, initially, we ranked the surrogate models based on their mean values and standard deviations. Then we selected the fist, middle and the last surrogate models (3



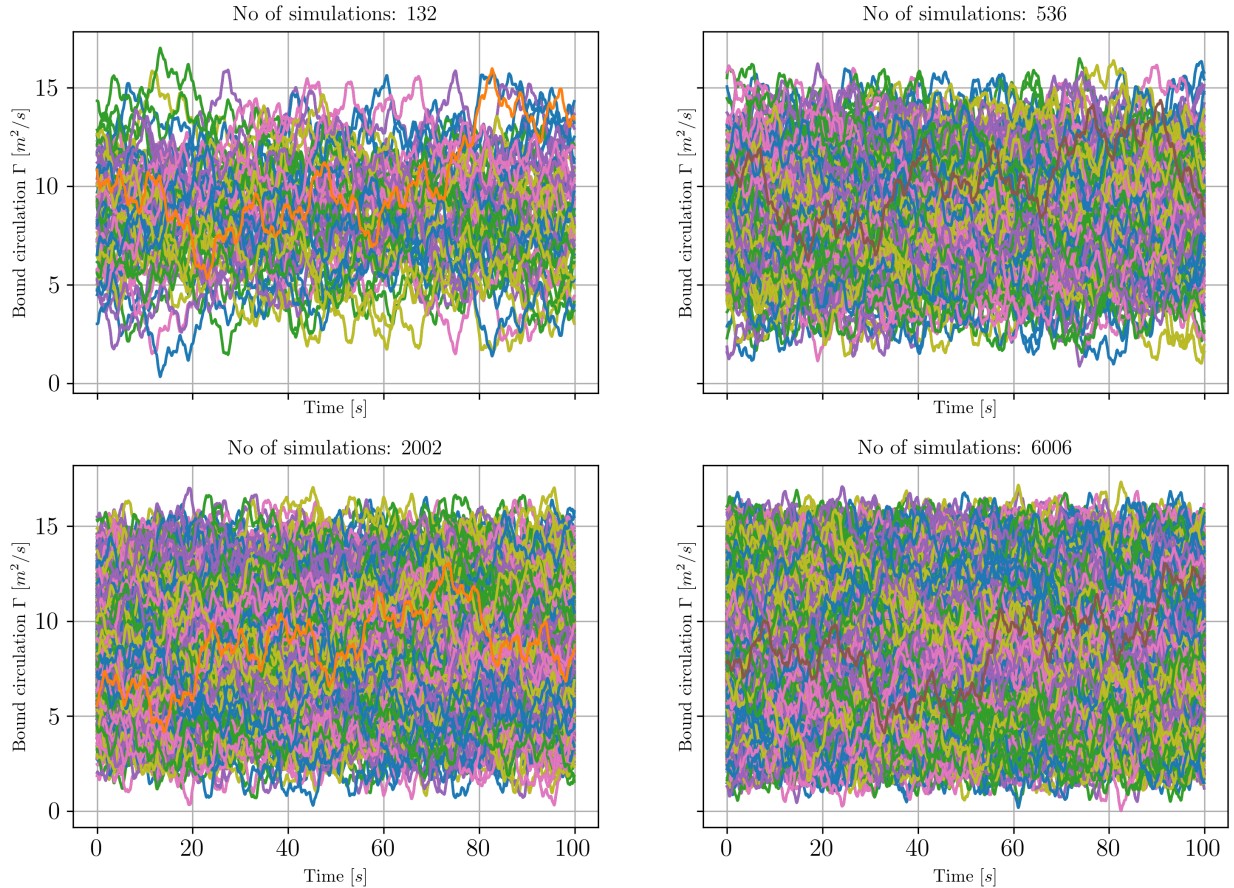

**Figure 6.** The first $100s$ of the time series used to build the surrogate models with polynomial degrees 2-5 respectively

surrogate model for each polynomial degree, in total 24 surrogate models) and took them thorough MCs $10^6$ times. Essentially, we took random samples from our 10 dimensional random domains for $10^6$ times and inserted those in the PCEs (Eq. (4)). Afterwards, we compare the histogram of those with one arbitrary time step of the reference case of 48000 simulations.

For each polynomial degree, regardless of the surrogate model location in the time series, the difference between the reference case and the MCs runs results change slightly. In other words, the difference between the MCs result histogram and the reference case histogram was only dependent on the polynomial degree, and not the position of the surrogate model in the time series as expected with the stationary process.

For the sake of space, we only show the MCs results for one set of four surrogate models. Figure 11 compares the histogram of 1 million MCs for the middle mean ranked surrogate model to the reference case at one arbitrary time step for four polynomial degrees.

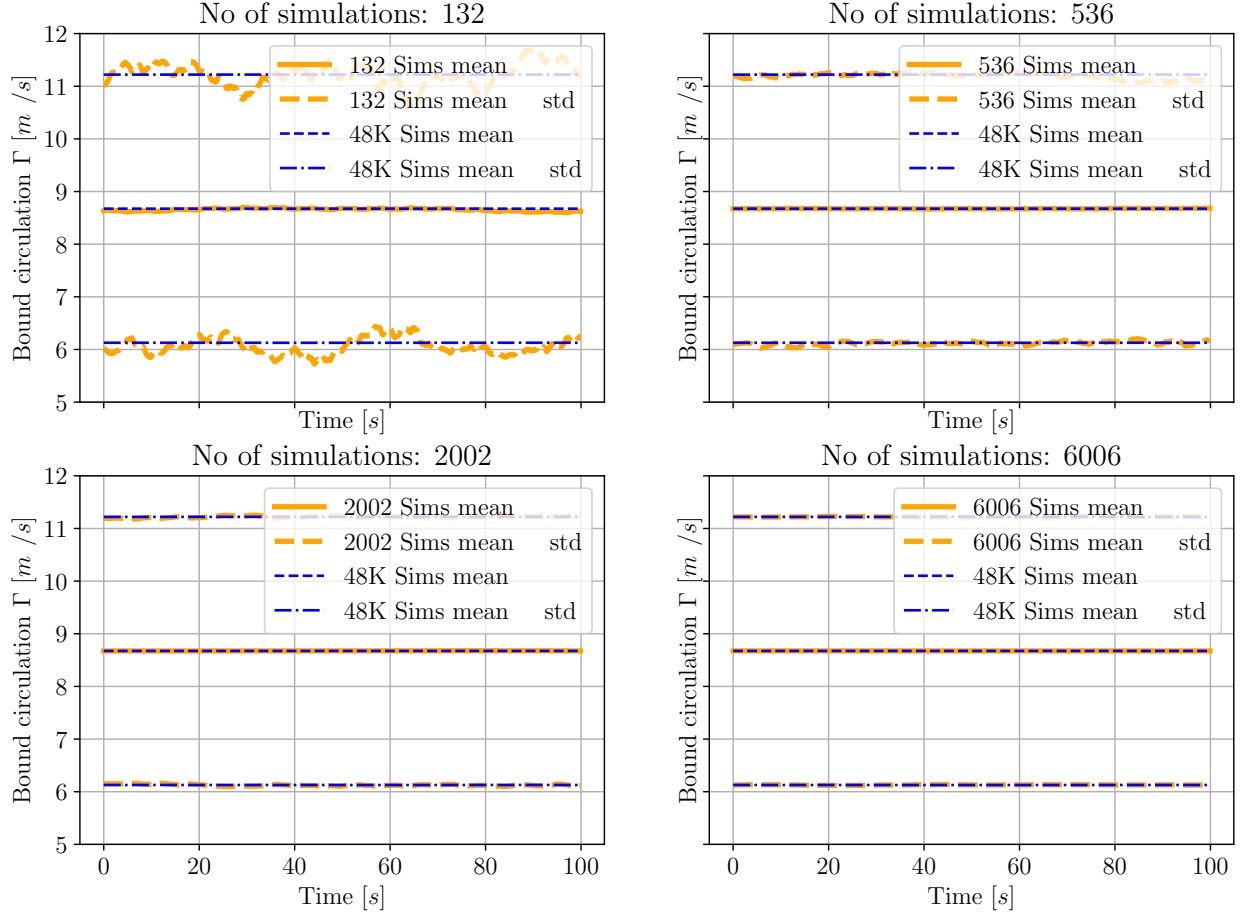

**Figure 7.** The first $100s$ of the time series mean and standard deviation bounds compared with the reference case

Figure 11 also shows the evaluation time for the $10^6$ MCs. As the polynomial degree increases, the evaluation time for the MCs increases. To give a sense of the efficiency of the results, one million time steps is almost equal to $167,600s$ simulations. As we have seen before, the extracted statistics from a small number of simulations do not represent the actual statistics and change enormously with progress through time. The reference case computation time was around $4$ hours. Therefore, almost $7.5$ minutes of emulation time to have statistical properties close to the reference case is justifiable. We performed all the simulations and emulations on the same computer. From a computational cost point of view, a combination of running $6006$ simulations for $2s$, building surrogate models on each time step, and then running the MCs for $10^6$ times, is cheaper than running $48000$ simulations in order to have smooth statistics.

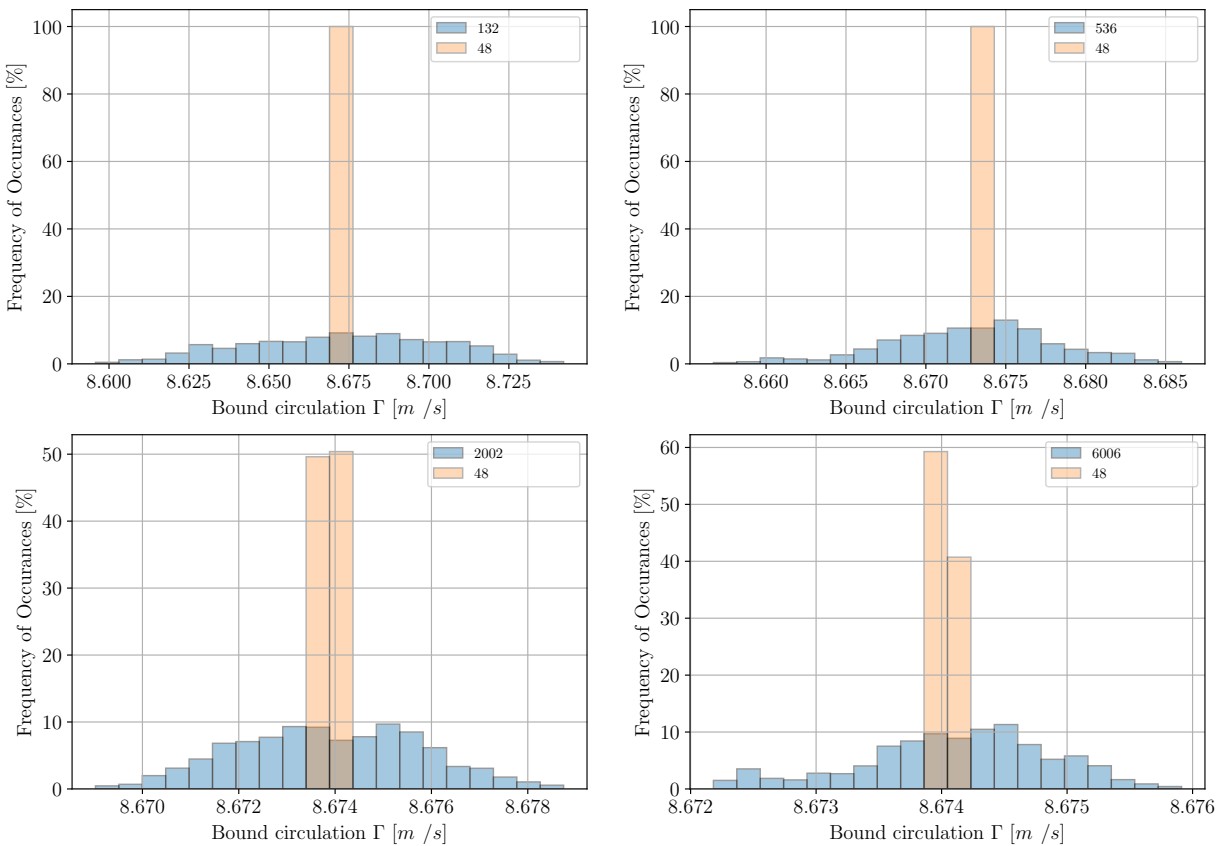

**Figure 8.** Mean values histogram compared with the reference case for 6000 time steps. The binning on the horizontal axis changes to accommodate the mean value range.

## 4 Conclusion

This paper is our initial attempt to build a surrogate model of time marching aerodynamic simulations. The form of the surrogate model that we used in this paper is a PCE. In Section 2.5, we explained the simple aerodynamic model used for this study. Also, we briefly described the method that we are using to build the PCEs. One major challenge with the building of the surrogate models is the *curse of dimensionality*, which we tried to tackle by using a reduced Veers model.

We showed how by increasing the number of simulations, the statistics of the results converge and do not change in time. As a result of this, building a few accurate surrogate models for a small length of time would suffice for our purpose. Therefore, to build an accurate surrogate model, we can reduce the simulation length significantly while increasing the number of simu-

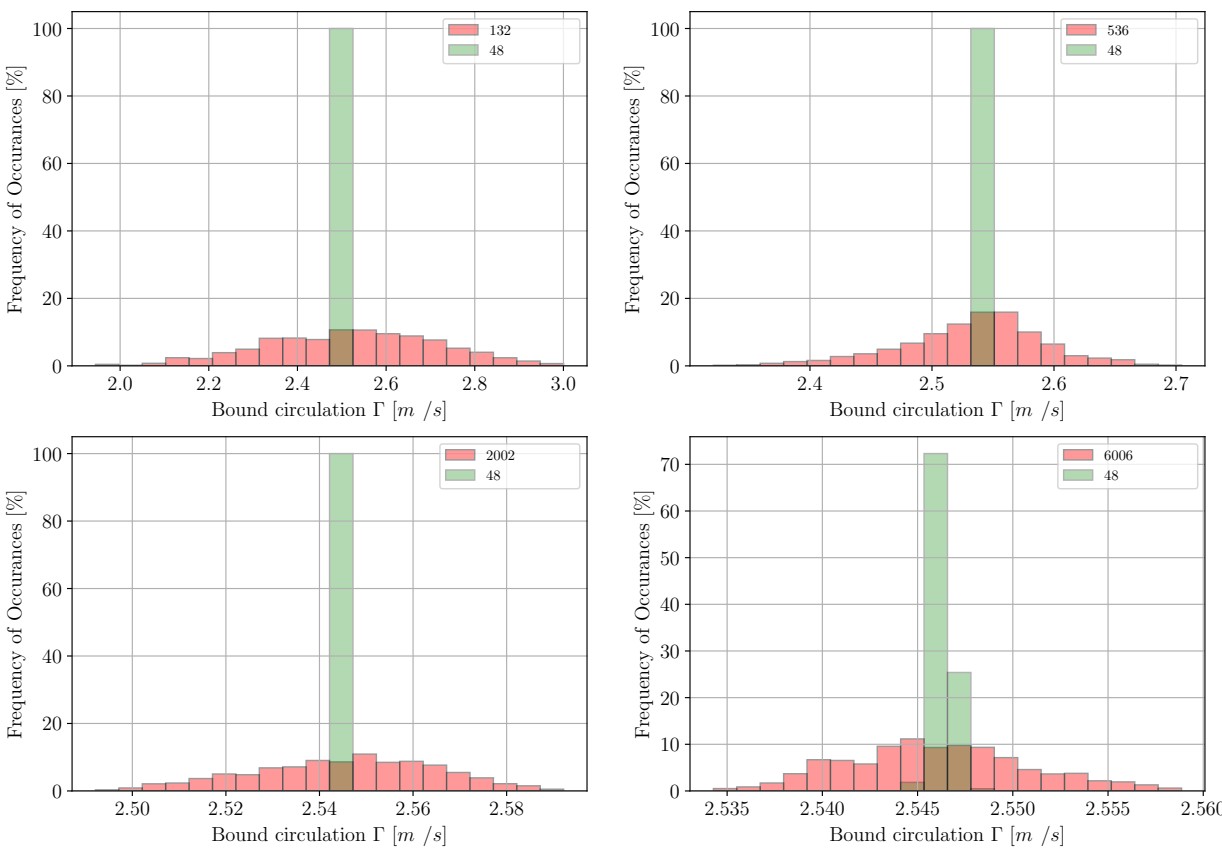

**Figure 9.** Standard deviation histogram compared with the reference case for 6000 time steps. The binning on the horizontal axis changes to accommodate the standard deviation value range.

lations. In the results section, we showed the surrogate model using a fifth-order polynomial built on 6006 simulations with a length of $2s$ gives us sufficiently accurate results in large MC runs to obtain output statistics.

The aerodynamic model is straightforward; therefore, the initial transient was less than 2 seconds (in fact $0.1s$ equal to one-time step). However, for future work, a smart way to deal with initialization time is essential; otherwise, increase the number of simulations would be very expensive. For example, if the required initialization time is $60s$, and we want to increase the number of simulations from 6 six hundred second simulations to 6006 two-second simulations, we are not doing any good in terms of computational cost. Aeroelastic and longer wakes will be studied for this challenge.

Another challenge is the practical application of this surrogate model. The surrogate model that we build in this study is one or a few time steps each efficiently the same due to stationarity. If we want to build a time series from this surrogate model, we have to sample the 10 dimensional random domain for the number of time steps to have a time series to post-process.





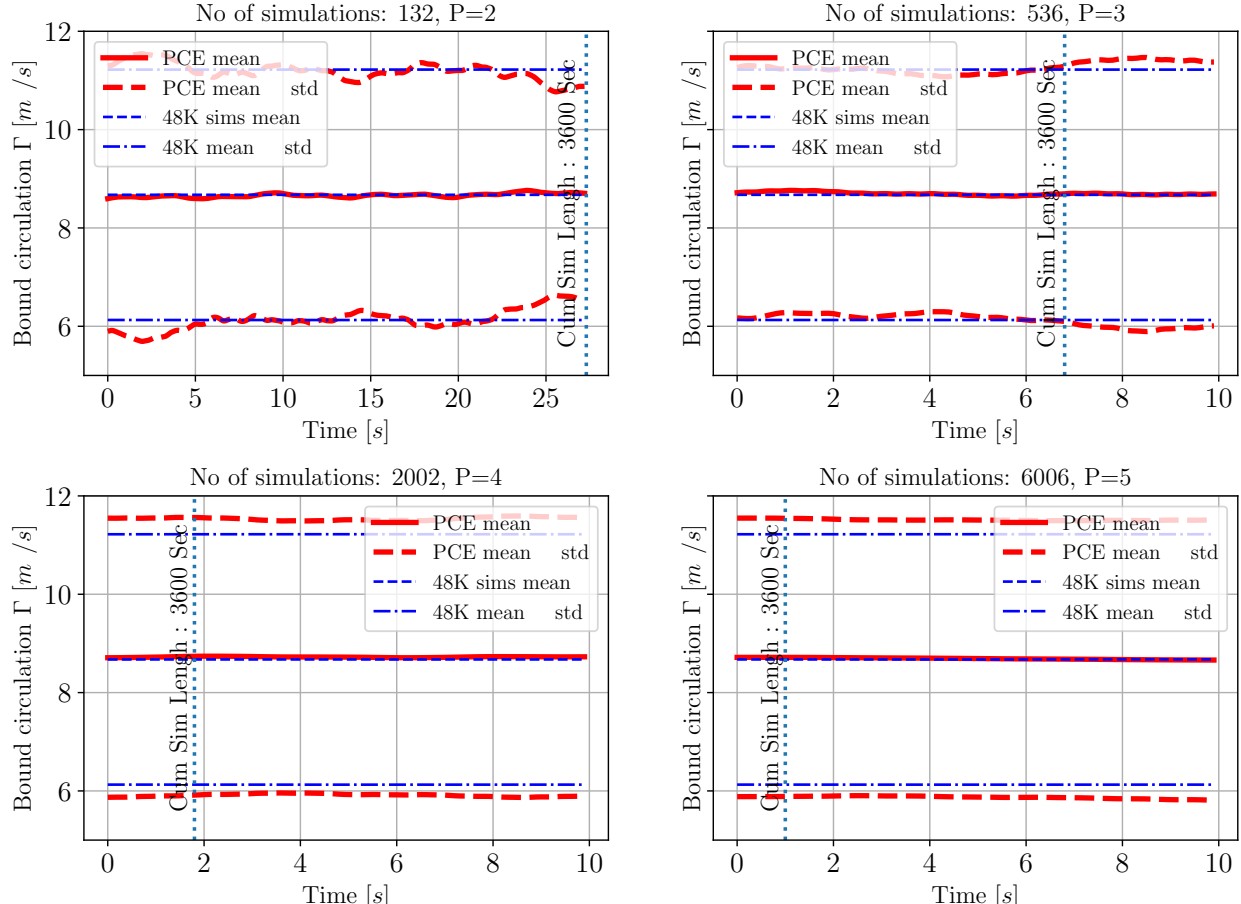

**Figure 10.** The mean and standard deviation comparison from the reference case ($48K$ simulations) and extracted values from PCEs. The number of simulations used to build the PCEs and polynomial degree $P$ are mentioned on the plots. The cumulative length of $3600s$ is shown with the vertical line.

However, the current method will miss the auto-correlation in the surrogate model result. This is important if we, for example, want to calculate fatigue loads from the surrogate model. This will require a surrogate form capable of resolving the correlation between time-steps. Fluck and Crawford did this previously for intrusive PCEs of an aerodynamic model.

It will be necessary to test the assumption that the sample statistical properties of the signal are not changing in time for the reference case against a nonlinear model. The current work uses a simple aerodynamic model, and therefore this assumption may not be valid if the model is a nonlinear or more complicated aeroelastic model.

For future work, we want to test the possibility of truncation of the polynomials in order to make the MCs of the surrogate model more cost-efficient. Also, using non-conventional polynomials, such as what Fluck and Crawford (2018) did, might result in a more efficient polynomial. In this study, we used the collocation method for calculating the PCE coefficients. There



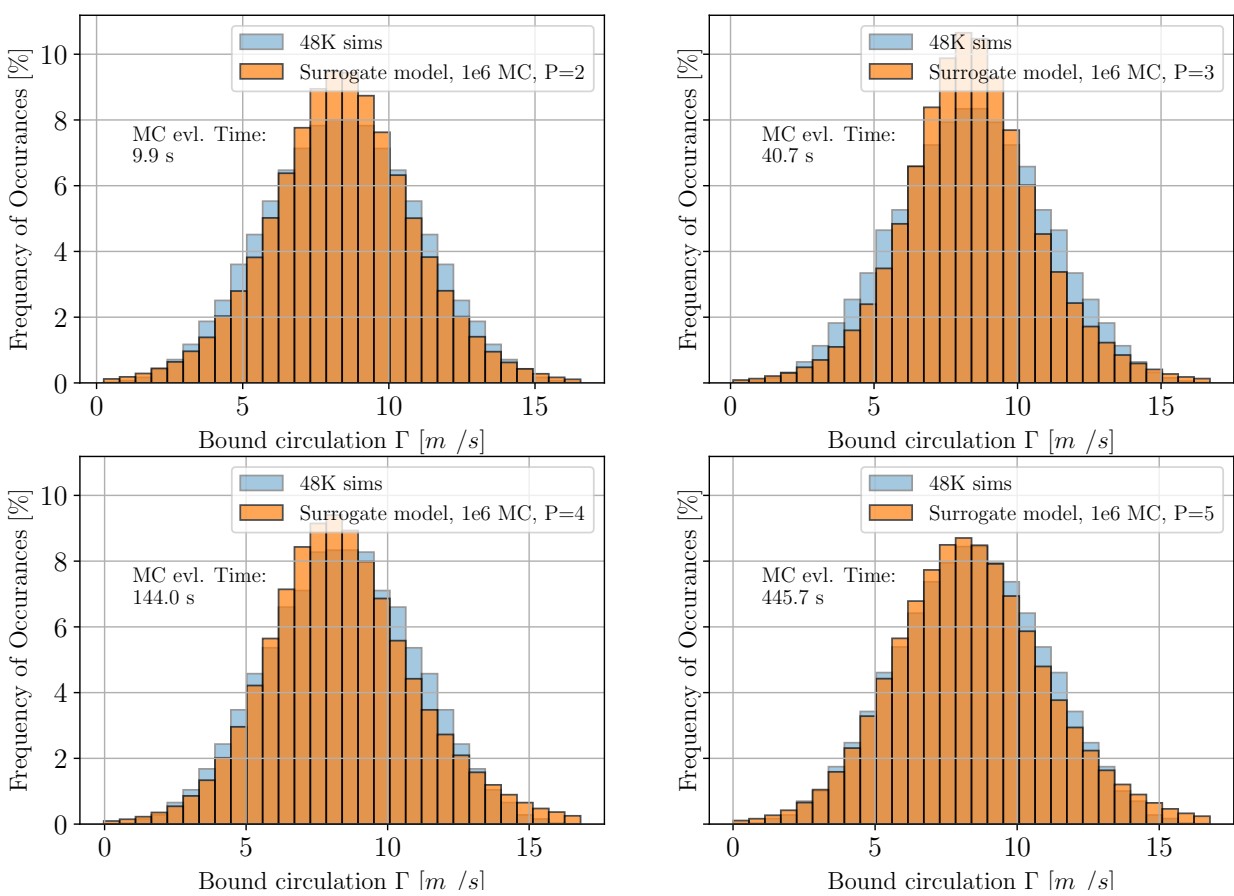

**Figure 11.** Surrogate model one million MCs vs the reference case

are more advanced quadrature and sparse methods (Sudret, 2007) that may need a smaller number of simulations at each time step using sparse techniques. Therefore, it is necessary to employ them in future studies. Finally, as mentioned before, this is an initial study with a simple aerodynamic model. In future work, we want to implement the method on a BEM or Lagrangian Vortex Models (LVM) and use commercial wind turbine simulation packages such as FAST to test the approach.

*Author contributions.* RH developed the necessary computer code and wrote the paper in consultation with and under the supervision of CC.

*Competing interests.* The authors confirm there are no competing interests are present.



*Acknowledgements.* We greatly acknowledge the funding for this study by the Natural Sciences and Engineering Research Council of Canada (NSERC).





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
