# Peer review of "Surrogate models for unsteady aerodynamics using non-intrusive Polynomial Chaos Expansions"

_Wind Energy Science, 2020_

## Referee Comment (RC1) · Erik Quaeghebeur (Referee) · 21 May 2020

**1   General comments**

**1.1   Summary of key points**

I have read the paper with interest. The topic of surrogate modeling for wind energy is an important one. The field has to deal with many complex, nonlinear models whose evaluation is computationally demanding. Having available techniques for building surrogates for such models that allow us to trade-off between accuracy and computational

efficiency, would be an enabler for new investigations and more advanced design. I support your efforts in this space.

The paper presents results on building surrogates for aerodynamic models relevant for load calculations. The surrogate modeling technique used, PCE, is established and popular in many domains. The test model used is linear and defined by a simple finite difference equation. Given an input process for the wind speed, defined as a Fourier series with random phases, the objective is to build a surrogate to estimate the mean and standard deviation of the output process.

The test model you chose is, I think, too simple. Namely, its linear, time-invariant nature, in combination with the Fourier series nature of the input, makes it possible to obtain explicit analytical expressions for the output. This means that building a surrogate from samples for this test problem makes no sense to me, as it can never be as efficient as evaluating the analytical expression. I provide the derivation of the analytical expressions in Sec. 2, to support my claims here.

In your paper, you consider this criticism and state:

> *We choose this simple aerodynamic model for the ease of simulations; hence, the simplicity of the aerodynamic model does not compromise the validity of the method.*

My reply to that is as follows:

- The conclusion does not follow from the premise: the simple model does not exhibit the difficult-to-capture behavior that a complex, nonlinear model has. Your method has not been tested to deal with such behavior and so you cannot convincingly claim it generalizes to those models.

- The specifics of your method include learning PCEs on shortened (in terms of time length) output sequences. However, you also establish the stationarity of
the output process. This practically means that time has no relevance anymore for the estimation of the output process statistics. Therefore, learning from a maximally shortened output, to one time instance, is enough. So the surrogate construction method you consider is too complex when dealing with output processes known to be stationary, even if the model is complex and nonlinear. So you would need to consider nonstationary processes, but for that again, you have no representative data to generalize from.

(Moreover, simulations are actually not necessary, as the output signals can be generated using analytical expressions.)

My conclusion, based on the single issue of using a test model that is too simple, is that the results you present do not advance the state of the art or provide insight not available more directly (analytically). *Therefore I advise the editors to reject this paper.* Furthermore, I do not think there are easy fixes: for surrogate modeling research, you need to demonstrate applicability to nonlinear systems that are similar to the ones used in practice. That would require new research and would lead to a different paper, not a rewritten version of this one.

I realize my judgment may feel harsh and of course other reviewers may disagree with me, but I feel I would be doing you a disservice in being less explicit and direct. I hope that the effort spent on this work allow you to more quickly take on and get results with alternatives.

I have further feedback, apart from the point focused on above. You can find an overview in the next subsection and detailed comments and questions in the attached annotated version of your paper.

**1.2   Overview of review criteria**

My judgments here are based on my current understanding of the work.

(Scale: Excellent, Good, Fair, Poor.)

**1.2.1 Principal criteria**

- SCIENTIFIC SIGNIFICANCE
  *Does the manuscript represent a substantial contribution to scientific progress within the scope of of WES (substantial new concepts, ideas, methods, analyses, or data)?*
  **Poor.** The authors have not convincingly argued that their surrogate creation method will generalize to realistic models; it is also not needed for the simple model they consider.

- SCIENTIFIC QUALITY
  *Are the scientific approach and applied methods valid? Is sufficient information given so other researchers (in principle) can repeat the work? Are the results discussed in an appropriate and balanced way (consideration of related work, including appropriate references)?*
  **Fair.** The information necessary for replication are not readily available. The discussion is insufficient to support the suggested conclusions. A critical look at the more general applicability of the method presented is missing.

- PRESENTATION QUALITY
  *Are the scientific results and conclusions presented in a clear, concise, and well-structured way (abstract conveys efficiently the essence of the paper; number and quality of figures/tables; appropriate, fluent, and precise use of English language)?*
  **Fair.** The mathematical development is insufficient and the many parts of the paper could be clarified and made more concise.

1.2.2 Specific aspects

1. *Does the paper address relevant scientific questions within the scope of WES?*
**Yes (excellent).** The topic of surrogate modeling of aerodynamic systems is clearly in scope for WES.

2. *Does the paper present novel concepts, ideas, tools, or data?*
**A few (fair).** The paper presents a method that is a combination of existing tools. There are a few novel aspects included (Hellinger distance, shortened learning sets for the surrogate), but their relevance and value is not convincingly established.

3. *Is the paper of broad international interest?*
**Yes (excellent).** The topic is generic and not restricted by location.

4. *Are clear objectives and/or hypotheses put forward?*
**Yes (good).** The abstract clearly states the goals (showing stationarity of the output process and building a surrogate to estimate statistics of that process). This could be done a bit more explicitly in the Introduction and the statistics in question could be mentioned.

5. *Are the scientific methods valid and clear outlined to be reproduced?*
**Partly (fair).** While a large part of the method can be deduced from the paper, not all aspects are described or described clearly enough.

6. *Are analyses and assumptions valid?*
**Not all (good).** A few assumptions are not valid (see annotated pdf).

7. *Are the presented results sufficient to support the interpretations and associated discussion?*
**No (poor).** The results pertain to a very specific, simple aerodynamic model.

The assumed generalizability to more complex, realistic models is not justified and doubtful according to my current understanding of the subject matter.

8. *Is the discussion relevant and backed up?*
**Limited (fair).** Once stationarity of the output process is established, much of the ensuing work becomes moot. So while the discussion afterwards is backed-up, I do not think it is really relevant.

9. *Are accurate conclusions reached based on the presented results and discussion?*
**Partly (fair).** The conclusions about the stationarity of the output process gloss over the transient too quickly and easily. (I think the authors should have the data to fix this.) The conclusions about the efficiency and therefore usefulness of the PCE surrogates are not clear enough: timings are not presented in a structured way and more efficient non-surrogate approaches to obtaining the results are not considered.

10. *Do the authors give proper credit to related and relevant work and clearly indicate their own original contribution?*
**Mostly (good).** The authors clearly indicate what work they build on, but their own contribution could be highlighted more explicitly.

11. *Does the title clearly reflect the contents of the paper and is it informative?*
**Not entirely (fair).** Based on the title, I had expected a method for creating surrogates for full outputs of a complex aerodynamic model, whereas the paper deals with a simplified, linear model and the surrogate is meant to deduce statistics only.

12. *Does the abstract provide a concise and complete summary, including quantitative results?*
**Limited (fair).** The bulk of the abstract sketches the context of the work. The

paper's results come in just two sentences at the end of the abstract. More of the paper's content could be summarized in the abstract.

13. *Is the overall presentation well structured?*
**Not in all aspects (fair).** The mathematical description of the method could be more fully developed and connected. Currently, the different techniques used are to a certain degree described, but in an isolated way. This approach hampers understanding when reading the paper.

14. *Is the paper written concisely and to the point?*
**No (poor).** There is a lot of repetition and also discussions that are vague due to a lack of mathematical formalization.

15. *Is the language fluent, precise, and grammatically correct?*
**Mostly (good).** Most of the text is correct English, but there were instances of mistakes (particles, some obvious grammar mistakes, some unclear sentences).

16. *Are the figures and tables useful and all necessary?*
**Partly (fair).** Many figures are not necessary or useful. (Once stationarity of the output process is established, plotting along a temporal axis becomes superfluous.)

17. *Are mathematical formulae, symbols, abbreviations, and units correctly defined and used according to the author guidelines?*
**Partly (fair).** Mathematical notation is mostly according to the guidelines, units are not. The definition of mathematical notation could be improved.

18. *Should any parts of the paper (text, formulae, figures, tables) be clarified, reduced, combined, or eliminated?*
**Yes (fair).** Please see the annotated pdf attached to this review.

[Figure]

19. *Are the number and quality of references appropriate?*
    **Yes (good).** It would be possible to give more references to sketch the wider context (e.g., https://www.wind-energ-sci.net/4/479/2019/), but that is not required, I think.

20. Is the amount and quality of supplementary material appropriate and of added value?
    **No supplementary material present (poor).** Code and generated data could have been made available. At least a justification should be given why such material is not made available.

**2 Specific comments**

As announced in Sec. 1.1, I here present a deduction of explicit, analytical expressions for the mean and variance of the output process. As part of this, an explicit expression for member signals of the output process is also derived.

The derivation below was obtained in limited time and has not been independently checked, so there may be things such as sign errors and off-by-one errors. However, I am quite confident the general shape of things is correct.

**2.1 The reduced Veers model is a truncated Fourier series**

In the text, no explicit expression for the random wind speed generated by the reduced Veers model is given. Actually, looking at the referenced paper (https://doi.org/10.1088/1742-6596/753/8/082009, Section 2.2), it becomes clear that it is a truncated Fourier

series with random phases:

$$u_{\infty,\xi}(t) = \begin{cases} 0, & t < t_0, \\ \bar{u} + \sum_{k=1}^{K} u_k \cos\left(\omega_k t + 2\pi\xi_k\right), & t \geq t_0, \end{cases}$$

where $\bar{u}$ is the mean wind speed, the $u_k$ are the other Fourier coefficients, $\omega_k$ are the frequencies considered, and $\xi_k$ is a random variable uniform over $[0,1]$. (The model also includes per-location phase offsets, which are not needed here, because only one location is considered.) This (type of) expression is very quick to give and so should be included in the paper, to clarify the model inputs.

For what follows, it is useful to have the expression for discrete time instances:

$$u_{\infty,\xi}(n) = \begin{cases} 0, & n < 0, \\ \bar{u} + \sum_{k=1}^{K} u_k \cos\left(\omega_k(t_0 + nT) + 2\pi\xi_k\right), & n \geq 0, \end{cases}$$

where $T$ is the time step. Because of the random phase components, we can equivalently define this as

$$u_{\infty,\xi}(n) = \begin{cases} 0, & n < 0, \\ \bar{u} + \sum_{k=1}^{K} u_k \cos\left(\omega_k nT + 2\pi\xi_k\right), & n \geq 0. \end{cases}$$

**2.2 The model considered is an LTI system**

In the text, the model considered is presented as a linear difference equation with constant coefficients:

$$\Gamma_{0,\xi}(n) - \kappa\Gamma_{0,\xi}(n-1) = \alpha u_{\infty,\xi}(n),$$

where $\kappa = \frac{C}{1-B}$ and $\alpha = \frac{A}{1-B}$. Therefore, it models an LTI system.

From LTI systems theory, we therefore know that its impulse response is

$$h(n) = \begin{cases} 0, & n < 0, \\ \alpha\kappa^n, & n \geq 0. \end{cases}$$

Note that $|\kappa| < 1$ is required for the system to be stable. We assume it is (based on the outputs I saw in the paper; this can be calculated from the parameters $B$ and $C$). The impulse response can be used to calculate any output value (for $n \geq 0$) using convolution:

$$\Gamma_{0,\xi}(n) = (u_{\infty,\xi} * h)(n) + \kappa^{n+1}\Gamma_0(-1).$$

Note that we need to know the initial condition $\Gamma_0(-1)$ to get the value of the transient at each time instant. In case $\Gamma_0(-1) = 0$, there is no transient.

The frequency response is:

$$H(\omega) = \sum_{n\in\mathbb{Z}} h(n)\exp(-j\omega n)$$
$$= \frac{\alpha}{1-\kappa\exp(-j\omega)}$$
$$= \frac{\alpha}{\sqrt{1-2\kappa\cos(\omega)+\kappa^2}} \exp\left(-j\overbrace{\tan^{-1}\left(\frac{\kappa\sin(\omega)}{1-\kappa\cos(\omega)}\right)}^{\phi_\kappa(\omega)}\right).$$

Because complex exponentials are eigenfunctions of LTI systems, we get an explicit expression for the convolution:

$$(u_{\infty,\xi} * h)(n) =$$
$$\frac{\alpha\bar{u}}{\sqrt{1-2\kappa+\kappa^2}} + \sum_{k=1}^{K} \frac{\alpha u_k}{\sqrt{1-2\kappa\cos(\omega_k T)+\kappa^2}}\cos\left(\omega_k nT + 2\pi\xi_k - \phi_\kappa(\omega_k T)\right).$$

This means that *we have an explicit, analytical expression for* $\Gamma_{0,\xi}(n)$.

**2.3  Process statistics**

Given the expressions for the inputs and outputs of the model, we can calculate their mean and variance. For this, we need to know the expectation of a cosine and squared

cosine with uniformly distributed argument:

$$\mathbb{E}\left(\cos(2\pi\xi_k + \ldots)\right) = \int_0^1 \cos(2\pi\xi_k + \ldots)\mathrm{d}\xi_k = 0,$$

$$\mathbb{E}\left(\cos^2(2\pi\xi_k + \ldots)\right) = \mathbb{E}\left(\frac{1 + \cos\left(2(2\pi\xi_k + \ldots)\right)}{2}\right) = \frac{1}{2}.$$

Then we get for the means:

$$\mathbb{E}(u_{\infty,\xi}(n)) = \bar{u},$$

$$\mathbb{E}(\Gamma_{0,\xi}(n)) = \frac{\alpha\bar{u}}{\sqrt{1 - 2\kappa + \kappa^2}} + \kappa^{n+1}\Gamma_0(-1).$$

So because of the transient (if there is a nonzero initial condition), the mean of the output process is not stationary (does depend on $n$).

For the variances (cross terms become zero because the $\xi_k$ are independent of each other):

$$\mathbb{E}\left((u_{\infty,\xi}(n) - \mathbb{E}(u_{\infty,\xi}(n)))^2\right) = \tfrac{1}{2}\sum_{k=1}^K u_k^2,$$

$$\mathbb{E}\left((\Gamma_{0,\xi}(n) - \mathbb{E}(\Gamma_{0,\xi}(n)))^2\right) = \tfrac{1}{2}\sum_{k=1}^K \frac{\alpha^2 u_k^2}{1 - 2\kappa\cos(\omega_k T) + \kappa^2}$$

The output process variance is stationary (does not depend on $n$).

**3  Other comments**

I provide various (other) comments in the annotated pdf of the paper provided in attachment. They include comments and suggestions about presentation and comments and questions that arose during my reading of the paper. Because of that, there

is overlap with what is discussed above. Nevertheless, I think it may provide value to
the authors to go over the comments, as many issues are not listed here and because
the context of the comments is obviously more apparent.

Please also note the supplement to this comment:
https://www.wind-energ-sci-discuss.net/wes-2020-24/wes-2020-24-RC1-
supplement.pdf

---

## Author Comment (AC1) · 4 Jun 2020

Dear Dr. Quaeghebeur,

Thanks a lot for your constructive comments. They are very helpful to improve my research in general and this journal paper specifically. We are working toward adding your analytical suggestion to the paper, and extend our study to a nonlinear model. We will update the journal paper in close future, after receiving and implementing other reviewers comments.

Looking forwards to see your comments on the next revision(s) of the paper.

[Figure]

Kind regards, Rad Haghi

---

## Referee Comment (RC2) · Anonymous Referee #2 · 16 Jun 2020

The paper presents an application of polynomial chaos expansions for uncertainty propagation through models of unsteady aerodynamics for wind energy purposes. In my opinion, this scientific topic is worth of being studied and be subject to publications, and the authors have made a fair effort in building a sound technical implementation of an uncertainty propagation algorithm. However, the current manuscript has several issues:

- Quality of descriptions and discussions. In a revised version of the paper, the presentation needs to be made much more thorough, with formal definitions of the variables, the steps in the analysis, model inputs and outputs, etc.

- Likewise, there are no quantitative results shown, and there is little discussion of the key findings of the paper, and how the results could be used further.

- The aerodynamic model is represented by a very simple one-step-memory autoregressive process, which is not necessarily capable of taking more complex dynamic effects into account (e.g. when using the one-step memory the process autocorrelation function is limited to an exponential decay function, hence harmonics and other dynamics due to structure motion can't be represented). This limits the usability of the conclusions such as e.g. the statements regarding using many short simulations rather than few longer ones. Some specific ideas of what could be improved are listed below. Given the amount of necessary changes, it would be most appropriate to resubmit the paper as a new manuscript rather than a revision of the current one.

**General comments**

1) The introduction lacks a proper description of what is considered an "aerodynamic model" (e.g. inputs, outputs, functionality) and what is its purpose. This is important in order to understand the motivation for the present study and define what the requirements for the surrogate model are.

2) I think you need to explain thoroughly what are the time and space dimensions used in the paper. We see a definition that the aerodynamic forces are a function of time and the incoming wind, but in the wind itself can be assumed both as a random process (in 1D), or as a random field with a certain coherence structure (in 3D), whereas the turbulence generation process normally makes use of the frozen turbulence hypothesis and a quasi-static wind field is generated, which is then advected with a predefined mean wind speed. Currently it is not clear which of these dimensions and properties are used and how are they included in the models.

3) Define clearly what is a "simulation" and what is a "wind time series"

4) Define the "surrogate model". What are the inputs and what are the outputs? How is time taken into account?

5) I would add a table with some comparison of the results for the different cases that have been run. Right now, Figures 7-10 show some qualitative results but there is no quantitative evidence and no clear recommendations as to what is the appropriate/recommended surrogate modelling approach for this problem.

6) What are the properties of the Veers model used? The wind field has autocorrelation which may influence the convergence of the simulation distributions.

7) The authors claim that "building a few accurate surrogate models for a small length of time would suffice for our purpose. Therefore, to build an accurate surrogate model, we can reduce the simulation length significantly while increasing the number of simulations". This is based on the experience from a one-step autoregressive model but how valid would this be in general?

8) What are the key findings of this paper? Please outline this clearly in the conclusions.

**Specific comments:**

9) Abstract: six seeds per condition is just a recommendation (a minimum requirement) from IEC61400-1, and not necessarily a common industrial practice.

10) Abstract: the description of the motivation for the present work is good, but the description of the present study and even more so, the outcomes of the study, are only vaguely defined. Please improve the abstract with more concrete descriptions of work done and results.

11) Page 1, lines 17-18: What is used is not exactly a Monte Carlo method (sampling from a specific distribution), it is more a sort of stratified sampling or importance sampling, since the simulations are done at predefined fixed wind speeds with numbers that do not correspond to a wind speed distribution.

12) Page 5, line 107: Isn't an autoregressive model with one-step memory also called a Markov chain?

13) Figure 6: what's the purpose of this figure? I think it can be left out without any loss of information.

**Technical comments:**

14) Abstract: typo – Balded written instead of Bladed

---

## Author Comment (AC2) · 16 Feb 2021

The authors would like to thank the reviewer for their time and greatly appreciate their feedback and suggestions to improve the article.

The article has gone through a major revision to address your and the other reviewers' comments.

- The quality of the descriptions and discussion has been improved. - The results part has been extended to cover your comment. - The aerodynamic model is updated to a more complex model that includes non-linearity.

The authors agree with your general and specific comments are addressed in the revised revision of the paper.

---

## Author Comment (AC3) · 16 Feb 2021

The authors would like to thank the reviewer for their time and greatly appreciate their constructive feedback and suggestions to improve the article. The article has gone through a major revision to include your comments and concerns. The authors generally agree with your feedback and used them to improve the manuscript.

- The aerodynamic model has been updated to a more complex model that includes the non-linearity - The analytical solution provided in your comments is added to the paper. - The results section is updated. - Your comments in the supplement PDF file have been taken into account.

[Figure]

The changes will appear in the revised version of the article.